# (MPO)²: Multivariate Polynomial Optimization based on Matrix Product Operators

## Abstract

Central to machine learning is the ability to perform universal function approximation and learn complex input-output relationships from limited numbers of observations. Suitable Multivariate Polynomial Optimization can in theory provide universal function approximations. However, the coefficients of the polynomial regression model grows exponentially in the polynomial degree. To reduce exponential growth tensor factorizations of the associated weight tensor have been explored, including the canonical polyadic decomposition (CPD) and tensor train (TT) decompositions. Whereas CPD has expressive power proportional to rank and current TT formulations are feature order dependent with each input feature associated to a specific factorization block. Furthermore, these procedures account for redundancies sub-optimally in the weight tensor. We presently explore multivariate polynomial optimization of matrix product operator (MPO) structures forming the (MPO)². Notably, the (MPO)² defines a flexible framework that naturally combines MPO polynomial weight tensors with MPO feature embeddings. The (MPO)² consequently produces an expressive yet compact representation of multivariate polynomials that is feature order independent and explicitly accounts for symmetries in the weight tensor. On a series of regression and classification problems we observe that the proposed (MPO)² provides superior performance when compared to existing tensor decomposition based multivariate polynomial regression approaches even outperforming conventional universal function approximation procedures on some datasets. The (MPO)² provides an expressive and versatile alternative to deep learning for universal function approximation with simple and efficient inference using second order methods.

## 1 Introduction

A central task in machine learning is to learn from data suitable functions that can map inputs to associated outputs in a way that best possible generalizes. Whereas it is well established that deep learning can provide universal function approximation for sufficiently large model architectures (Hornik et al., 1989), many other modeling tools exist for universal function approximations. This includes Gaussian Processes (GPs) for suitable choices of kernels (Williams & Rasmussen, 2006; Tran et al., 2016) and polynomial regressions, i.e. as also guaranteed by Taylor's theorem.

In the recent decade context aware learning methods have demonstrated superior performance in generalization enabling to leverage nonlinear dependencies. This includes the transformer architecture (Vaswani et al., 2017) and gating mechanisms as used for instance in long-short term memory (Hochreiter & Schmidhuber, 1997) and gated linear units (Dauphin et al., 2017). Importantly, such architectures can directly leverage multiplicative interactions between attributes instrumental in deep learning (Jayakumar et al., 2020). Whereas multiplicative interactions are well established in classical statistics as defined by widely used interaction terms between features, the ability to directly account for simple multiplication operations within a standard multilayer feed forward network has been shown to require several (i.e., four) hidden neurons (Lin et al., 2017).

Conversely, polynomial networks directly express multiplicative interactions using higher degree terms. Early works on polynomial networks were based on the pi-sigma network (Shin & Ghosh, 1991) that expressed polynomials as multiplications of simple linear regression functions. Ridge polynomial networks (Shin & Ghosh, 1995) similarly express the polynomial function in terms of successive accumulations whereas the pi-sigma-pi network introduced an additional multiplicative layer combining multiple pi-sigma networks (Li, 2003).

Recently, polynomial networks have been advanced exploring tensor decomposition structures including the canonical polyadic decomposition (Hendrikx et al., 2019; Govindarajan et al., 2022; Ayvaz & De Lathauwer, 2022; Kilic & Batselier, 2025) as well as hierarchically coupled CPD decompositions forming the P-Net (Chrysos et al., 2022b;a) which are directly related respectively to the pi-sigma and ridge polynomial networks that can be considered rank one CPD structured. Besides the CPD, these approaches have also been advanced to more flexible tensor network structures including the tensor train/matrix product states decomposition (MPS/TT) (Stoudenmire & Schwab, 2016b; Götte et al., 2021; Kilic & Batselier, 2025). Importantly, decomposed polynomial networks can be optimized using simple alternating linear systems (ALS) optimization using second order methods to optimize each factor of the decomposition at a time, see also Hendrikx et al. (2019); Ayvaz & De Lathauwer (2022); Kilic & Batselier (2025). Whereas the above polynomial networks explore tensor decompositions for regression we note that they differ from tensor regression which aims to explore regression of high order data structures (Liu et al., 2022). Tensor network representation for polynomial networks also differ from recent efforts using tensor decomposition procedures to compress the weight tensors in deep learning models, for a discussion of the connections between tensor decompositions and deep learning see also Panagakis et al. (2021).

Importantly, tensor decomposition structures address the curse of dimensionality of the multivariate polynomial regression weights (Shin & Ghosh, 1991). However, the existing formulations using the CPD decomposition have limited expressive power, whereas the current MPS/TT based modeling procedures are feature order dependent. Furthermore, these procedures do not account for redundancies in the weight tensor and relies on prespecified feature representations. These limitations, we argue, have hampered the wider adoption of this otherwise attractive alternative to deep learning based function approximation.

We presently propose the Multivariate Polynomial Optimization based on Matrix Product Operators $(MPO)^2$ framework, a new tensor network based structure for the modeling of higher order polynomials. Notably, $(MPO)^2$ generalizes polynomial tensor networks enhancing their

- **Expressiveness:** We consider more expressive tensor network representations exploring the matrix product operators formalism to both learn feature and polynomial representations with added expressive capabilities when compared to CPD and existing MPS/TT based procedures notably also being feature order independent when compared to the latter.

- **Versatility:** We introduce generic structured operators to account for inductive biases such as polynomial degree redundancies and translation invariance as imposed by conventional convolutional neural networks. We further accommodate different loss functions such as least squares for regression and cross-entropy minimization for classification using a loss agnostic second order minimization framework.

We evaluate the proposed $(MPO)^2$ structure for supervised learning on several tabular datasets and on image classification, and highlight its advantages over the latest tensor network based methods.

## 2 Methods

### 2.1 Tensor Networks and Tensor Notation

Tensor networks are represented by graphs where each node is a tensor and the connections represent contraction over a mode of the tensor.

In this work, the position of the indices of a tensor, when referring to tensor networks structures, will be at the apex to indicate vertical modes in the diagrammatic representation and at the pedex to indicate horizontal modes. The different positions have no mathematical difference, but carries a conceptual difference where vertical modes will be associated to the input space while horizontal modes will be associated with the latent space. Summation over multiple indices of the tensor $\mathsf{M}$ with elements $M_{i_1 i_2 \dots i_n}$ will be denoted by $\sum_{i_1 i_2 \dots i_n} M_{i_1 i_2 \dots i_n} = \sum_{\boldsymbol{i}} M_{\boldsymbol{i}^{(n)}}$, where $\boldsymbol{i}^{(n)} = \{i_1, i_2, \dots, i_n\}$, meaning that the sum is performed over all the indices going from $i_1$ to $i_n$. When the apex is omitted, it means that $\boldsymbol{i} = \boldsymbol{i}^{(N)}$, where $N$ is the degree of the polynomial.

A trivial example of a graphical representation of tensor networks can be seen in Figure 1 in which, in the left panel, a tensor with five modes is illustrated, and in the middle panel a contraction of two tensors multiplied along one mode corresponding to conventional matrix multiplication, and in the right panel the matrix product operator (MPO) corresponding to multiple tensors being pairwise contracted along one mode.

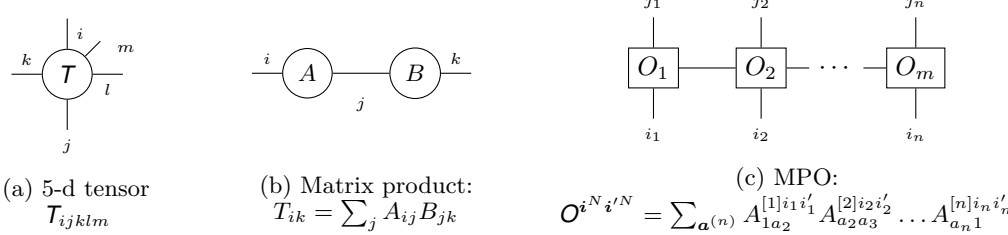

(a) 5-d tensor
$T_{ijklm}$

(b) Matrix product:
$T_{ik} = \sum_j A_{ij} B_{jk}$

(c) MPO:
$O^{\boldsymbol{i}^N \boldsymbol{i}'^N} = \sum_{\boldsymbol{a}^{(n)}} A_{1a_2}^{[1]i_1 i_1'} A_{a_2 a_3}^{[2]i_2 i_2'} \dots A_{a_n 1}^{[n]i_n i_n'}$

Figure 1: Graphical representation of (a) tensors, (b) the matrix product, and (c) the matrix product operator (MPO).

The well-known tensor network structures, namely *matrix product states* (*MPS*) and *tensor trains* (*TT*), as well as Tucker and CPD decompositions, are respectively given by

$$\text{MPS/TT:} \quad T_{\boldsymbol{d}^{(N)}l} = \sum_{\boldsymbol{r}}^{R} T_{1r_2}^{[1]d_1} T_{r_2 r_3}^{[2]d_2} \dots T_{r_n 1}^{[n]d_n l} \quad (1)$$

$$\text{Tucker/CPD:} \quad T_{\boldsymbol{d}^{(N)}l} = \sum_{\boldsymbol{r}}^{R} \mathcal{G}_{\boldsymbol{r}} T_{r_1}^{[1]d_1} \dots T_{r_n}^{[n]d_n}, \quad (2)$$

in which the Tucker decomposition reduces to the CPD when $\mathcal{G} = \mathcal{I}$, i.e., is defined as the identity tensor with ones along the (hyper-)diagonal and zeros elsewhere. Notably, these decompositions are special cases of MPOs.

## 2.2 Multivariate Polynomial Regression

Given an input $\boldsymbol{x}$ of dimension $D$, we define a multivariate polynomial of $\boldsymbol{x}$ of degree $N$:

$$p_l(\boldsymbol{x}) = T_l + \sum_{d_1} T_{l d_1} x_{d_1} + \sum_{d_2 \geq d_1} T_{l \boldsymbol{d}^{(2)}} x_{d_1} x_{d_2} + \dots + \sum_{d_2 \geq d_1} \dots \sum_{d_N \geq d_{N-1}} T_{l \boldsymbol{d}^{(N)}} x_{d_1} \dots x_{d_N} \quad (3)$$

where $l$ indicates one of the multivariate polynomial outputs, and $T$ are the coefficients.

Following Ayvaz & De Lathauwer (2022), we consider two formulations of the polynomial. Namely, as a sum of independent homogeneous polynomials of increasing degree with a tensor for each degree specifying all coefficients within this degree (type I), or one tensor to represent all coefficients of the polynomial (type II):

$$\text{Type I: } p_l(\boldsymbol{x}) = \sum_{n=0}^{N} \sum_{\boldsymbol{d}^{(n)}} T_{l \boldsymbol{d}^{(n)}} x_{d_1} \dots x_{d_n}, \quad \text{Type II: } p_l(\tilde{\boldsymbol{x}}) = \sum_{\boldsymbol{d}^{(N)}} \tilde{T}_{l \boldsymbol{d}^{(N)}} \tilde{x}_{d_1} \dots \tilde{x}_{d_N} \quad (4)$$

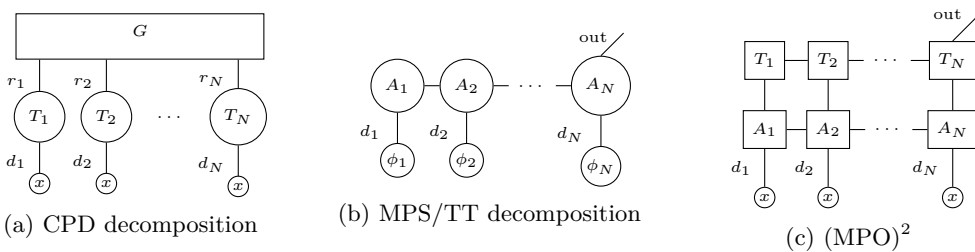

(a) CPD decomposition    (b) MPS/TT decomposition    (c) $(\text{MPO})^2$

Figure 2: Existing tensor network modeling procedures for multivariate polynomial regression based on (a) the CPD decomposition, (b) the MPS/TT decomposition in which each feature has its own train-cart operating on a fixed transformation $\phi(x_i) = \phi_i$ imposed on each feature, and (c) the proposed $(\text{MPO})^2$ framework exploring two layers of MPOs respectively transforming the input to suitable latent representations and creating a feature order invariant polynomial representation.

where $\tilde{\boldsymbol{x}}$ will be the input vector $\boldsymbol{x}$ with a constant additional feature of value 1 in order to account for all coefficients of all the different degrees in the modeling.

Notably, these weight tensors grow exponentially in the number of coefficients as the degree $N$ of the polynomial increases for M features by $\mathcal{O}(\mathcal{M}^{\mathcal{N}})$ making the polynomial regression considering all coefficients of all degrees infeasible except for low degree polynomials with limited numbers of features. To reduce the number of parameters these weight tensors have been decomposed considering the CPD decomposition (Hendrikx et al., 2019; Ayvaz & De Lathauwer, 2022; Govindarajan et al., 2022; Chrysos et al., 2022a; Kilic & Batselier, 2025) as well as tensor train decomposition (Stoudenmire & Schwab, 2016b; Götte et al., 2021; Kilic & Batselier, 2025). However, these existing CPD procedures have limited modeling capacity whereas the existing TT modeling procedures are feature order dependent decomposing the weight tensors using feature specific carts, i.e., $\mathcal{T}^{[m]d'_{m-1}}_{r_{m-1}r_m}$ associates the decomposed weight tensor with feature $m$, which is undesirable as there typically exists no natural feature ordering of the decomposition. As we will show, these drawbacks can be effectively addressed considering the MPO formalism.

### 2.2.1 $(\text{MPO})^2$: Multivariate Polynomial optimization using Matrix Product Operators

Often in machine learning, to enhance the capability of the model, a linear transformation is applied to the inputs to learn suitable latent feature representations. By applying a generic set of transformations $\boldsymbol{A}^{[i]}$ to the inputs, we can express the polynomial as:

$$p_l(\boldsymbol{x}) = \sum_{\boldsymbol{dd'}} \mathcal{T}_{\boldsymbol{d'}l} A^{[1]}_{d'_1 d_1} x_{d_1} \dots A^{[N]}_{d'_N d_N} x_{d_N} = \sum_{\boldsymbol{dd'}} \mathcal{T}_{\boldsymbol{d'}l} \mathbf{A}_{\boldsymbol{d'd}} x_{d_1} \dots x_{d_N} \tag{5}$$

where we omit the apex $\boldsymbol{d}^{(N)}$ when it is equal to $N$, the degree of the polynomial. We defined the product of all linear operators $\boldsymbol{A}$ as a tensor $\mathbf{A}$.

In the $(\text{MPO})^2$ framework we propose to perform multivariate polynomial regression and classification by modeling both the generic linear transformation of the input space as well as the polynomial coefficients as *matrix product operators* (MPOs). These are diagrammatically represented in Figure 2c and given as follows:

$$\mathcal{T}_{\boldsymbol{d'}^{(N)}l} = \sum_{\boldsymbol{r}}^{R} \mathcal{T}^{[1]d'_1}_{1r_2} \mathcal{T}^{[2]d'_2}_{r_2 r_3} \dots \mathcal{T}^{[n]d'_n l}_{r_n 1}, \quad \mathbf{A}_{\boldsymbol{d'd}} = \sum_{\boldsymbol{r}}^{R} A^{[1]d_1 d'_1}_{1r_2} A^{[2]d_2 d'_2}_{r_2 r_3} \dots A^{[n]d_n d'_n}_{r_n 1} \tag{6}$$

where $R$ is the rank of the MPO structure.

We propose three structures of the MPOs representing the input transformation tensor **A** by linear projections, convolutions, and masking that accounts for weight redundancies by the proposed masking MPO. These three structures are outlined below, whereas a more comprehensive mathematical description can be found in appendix A.1.

### 2.2.2 LINEAR PROJECTION MPO

We project the inputs subspace, related to a single degree, from $D$ to $D'$ dimensional space, where $D' \ll D$, reducing parameters and complexity of inverting Hessian during inference (see section 2.3) by a factor $\sim \left(\frac{D}{D'}\right)^3$. The operator in its most general form as in equation 8 can be randomly initialized and learned blockwise in the same fashion as the train structured coefficients are learned, in such a way that the model learns the best transformation of the inputs. Especially for high-dimensional inputs, computing and inverting the Hessian of a block can be challenging, and, if the inputs show linear dependency, wasteful. We introduce a learnable operator that applies a simple linear transformation for each subspace represented by the blocks, which results in global structured linear transformation. To further reduce the parameters, we can impose independency between the subspaces for the linear transformation by setting the rank to one. The advantage is that the new model, instead of representing the coefficients of the polynomial with blocks of dimension $R^2 D$, instead is represented by two blocks of dimensions $R^2 D'$ and $DD'$, where $D$ is the dimension of the input and $D'$ is the dimension of the projected subspace. We define the linear MPO block as a randomly initialized learnable tensor $\boldsymbol{B}_{r_i r_{i+1}}^{i_i i'_i}$. When the dimension of the rank $r$ is 1 we retrieve linear independent transformations of the inputs.

### 2.2.3 CONVOLUTION MPO

A structured case of linear projection are convolutions, which accounts for translation invariant compression as explored in CNNs. By representing the inputs as a two dimensional tensor, and project the inputs along one of these two dimensions we obtain an operation equivalent to convolution. For images, we can define the two dimensions as patches and pixels in each patch (Olshausen & Field, 1996), respectively called $p$ and $k$ in the following. The convolution with kernel $g$ can then be written has $x_p = \sum_k g_k X_{k,p}$. Consequently, we define the convolution MPO block by $G^{(k_n, p_n)p'_n} = g_{k_n} \delta_{p_n p'_n}$ where the index $(k, p)$ represents one index obtained by vectorizing over the dimension in the parentheses, where $\delta$ is the delta function, taking value 1 only if all indexes are the same and 0 otherwise. As a result, the MPO convolution block is defined as a linear projection on a subset of the full space. Notably, using the MPO formalism it is natural to also increase the multiplicity of the kernels by increasing the MPO's rank dimension, defining the MPO convolution blocks as $\boldsymbol{G}_{a_n a_{n+1}}^{(k_n, p_n)p'_n} = \boldsymbol{G}_{a_n a_{n+1}}^{k_n} \delta_{p_n p'_n}$, such that the kernels over different subspaces interact.

### 2.2.4 MASKING MPO

The existing tensor network based polynomial regression procedures have degenerate polynomial coefficients as defined in equation 3 in which the weight tensor includes all orderings of multiplications of the same terms. Imposing symmetric constraint on the coefficients, is often hard to model, especially for tensor decomposition methods. For this reason in the modeling of polynomial the symmetry is disregarded, leading to a number of represented parameters that in the worst case scales as depending on the model specifications. Modeling the degenerate polynomial in equation 4 respect to equation 3, increases the coefficients to be modeled of a factor of $K \simeq e^{-n} n^n$ (which for $n = 6$ is $\simeq 115$ and for $n = 10$ it is $\sim 5 \times 10^4$). For large polynomial degrees $n$, the divergence of $K$ can impair the expression power and the explainability of the model. The ideal scenario, would be to associate each combination of input (monomial) to one and only one element of the model output. We can achieve this by introducing a mask that filters the tensor of coefficients obtaining the model for the polynomial defined in equation 3. We model the mask filtering degenerate coefficients as an MPO by defining its blocks **C** as $\boldsymbol{C}_{b_i b_{i+1}}^{i_i i'_i} = \sum_k H_{b_i k} \boldsymbol{D}_{k i_i i'_i b_{i+1}}$, where $H_{ij} = \theta(j - i)$, $\boldsymbol{D}_{ab...} = \delta_{ab...}$ and $\theta$ is the Heaviside function. The polynomial can consequently, be expressed as a contraction between a tensor representing coefficients and a masking MPO and a tensor

containing the inputs, all retaining the block structure. Notably, due to this separate masking MPO, the gradient and subsequent Hessian calculations used for inference (see section 2.3) remain unchanged.

### 2.3 EFFICIENT MODEL INFERENCE

In multivariate polynomial optimization, CPD has parameterized polynomial coefficients and enabled efficient Gaus-Newton (GN) implementations defining the TeMPO procedure for least squares minimization (Ayvaz & Lathauwer, 2021). For MPS/TT and MPO, traditionally training proceeds by alternatively solving subproblems defined per block of the tensor network utilizing alternating linear systems (ALS) and density matrix renormalization group (DMRG) update schemes (Holtz et al., 2012; Grasedyck et al., 2019). Using these methodologies MPS/TT models have demonstrated competitive supervised learning performance (Stoudenmire & Schwab, 2016a). Leveraging these prior works, we adopt the efficiency of natural gradient based ALS within our $(MPO)^2$ framework. To our knowledge, no prior work jointly tackles *MPO structured multivariate polynomials*, *linear, convolutional and masking operators*, and *alternating natural gradient* training. Our formulation closes this gap, unifying tensorized polynomial modeling with loss agnostic optimization.

#### 2.3.1 ALTERNATING NATURAL GRADIENT

To learn the block of the MPOs we use a block-wise version of the natural gradient method, optimal for large family of Bregman divergence losses, incl., neg. log likelihood, cross entropy (classification) and least squares (regression), see Amari (2016).

Notably, standard gradient updates when dealing with MPS/TT have been demonstrated to lead to slow convergence (Qiu et al., 2024). The natural gradient is a second order optimization method, that calculates the update step of the parameters taking in consideration the curvature or the loss, resulting to faster learning. Often utilized algorithms for MPOs are alternating least squares or the density matrix renormalization group, both sharing similar properties and methods.

Given an objective loss to minimize $min_\theta L(y, x(\theta))$, natural gradient defines the best update of the parameters as $\Delta\theta = -\boldsymbol{H}_\theta^{-1}(L)\boldsymbol{j}_\theta(L)$, where $\boldsymbol{H}_\theta(L)$ and $\boldsymbol{j}_\theta(L)$ are the Hessian and the Jacobian of the loss respect to the parameters. Inspired by alternating least squares methodologies on tensor networks, we learn the update step block-wise. The method reduces to computing the Hessian and Jacobian of the loss with respect to a block, updates the block according to the step, and repeats the process until all blocks are updated and then proceeds to repeat the operation in the opposite direction. We denote the full iteration as a *swipe*.

The Hessian is often singular in the first swipes, especially when considering other losses than least squares minimization, due to random initialization. To stabilize the inference, we introduce Tikhonov regularization (Boyd & Ong, 2009; Trefethen, 2019), with an exponentially decaying schedule for weight decay.

Importantly, for MPO structures, calculating the Hessian of a single block reduces to a trivial task. We can write the Hessian and Jacobian taking into account the regularization:

$$\boldsymbol{j}_{\mathbf{A}^{(i)}}(L) = \sum_s \sum_l \nabla_{\mathbf{A}^{(i)}} p_{ls}\ \partial_{p_{ls}} L(\boldsymbol{p}_s, y_s) + \lambda\mathbf{A}^{(i)}$$

$$\boldsymbol{H}_{\mathbf{A}^{(i)}}(L) = \sum_s \sum_{ll'} \nabla_{\mathbf{A}^{(i)}} p_{ls}\ \nabla_{\mathbf{A}^{(i)}}^T p_{l's}\ \partial_{p_{ls}}\partial_{p_{l's}} L(\boldsymbol{p}_s, \boldsymbol{y}) + \lambda\boldsymbol{I} \tag{7}$$

where $p_{ls}$ is the output of the model for sample $s$ and output dimension $l$ and $\boldsymbol{p}_s$ is the vector of outputs for sample $s$.

Block-wise learning and MPO structured coefficients simplify the Hessian since $\nabla_{\mathbf{A}^{(i)}}\nabla_{\mathbf{A}^{(i)}} p = 0$. Additionally, the gradient with respect to a block, amounts to calculating the contraction of the full MPO without the differentiated block.

Notably when using the least squares loss the natural gradient method is equivalent to the more commonly used ALS method defined in Holtz et al. (2012).

## 3 Related Works

Using tensor products to model general "feature-output" maps has a long history going back to Pi-Sigma networks (Shin & Ghosh, 1991). Recently, higher order Sigma–Pi and Sigma–Pi–Sigma neural networks (SPSNNs) were proposed (Jiao & Su, 2024; Deng et al., 2024; Sarıkaya et al., 2023). They use multiplicative units in place of neurons and parameter sharing over layers to compactly encode polynomial maps.

Theoretically, multiplicative networks can approximate smooth targets with fewer layers/neurons than ReLU nets (Ben-Shaul et al., 2023; Jayakumar et al., 2020), while biologically inspired multiplicative couplings accelerate learning and gating in RNNs (Zhang et al., 2025). Deep polynomial networks such as Π-nets combine hierarchical polynomial expansions with CPD (rank 1) implementations and parameter sharing across layers to curb parameter growth and achieve strong results in vision (Chrysos et al., 2022b).

Tensor Machines learn target–specific polynomial features via low–rank CPD tensors (Yang & Gittens, 2015). These typically assume CP/Tucker parameterizations and squared/logistic losses. From input output mode perspective, they can be seen as a variants of multivariate polynomial models of recent work (Ayvaz & Lathauwer, 2021).

As opposed to rank-1 models, our approach leverages the exponentially higher expressivity of MPO over CPD, as shown in Oseledets (2011), while avoiding the quadratic growth in block parameters ($\approx R^2$ for MPO versus $\approx R$ for CP). Our MPO model generalizes aforementioned Π-nets without layer nonlinear activations and multivariate polynomial models by offering unifying architecture based on arbitrary rank decompositions and multilinear filters, such as convolution or (random) feature projections (Kar & Karnick, 2012).

## 4 Results and Discussion

We present a comparison between $(\text{MPO})^2$, CPD for polynomial regression using symmetric CPD (CPD-S) based on TeMPO (Ayvaz & De Lathauwer, 2022) and asymmetric CPD (CPD-A) (Govindarajan et al., 2022) optimized in our framework. We further include the classical TT/MPS structure for regression both with Fourier basis (TNML-F) as in Efthymiou et al. (2019) and polynomial basis (TNML-P) as in Götte et al. (2021). The TNML-F models are optimized using the present optimization framework to directly assess the impact on model structure on performance whereas the original paper utilizes density matrix renormalization group (DMRG). For comparison, we also included Gaussian Processes (GP) and XGBoost as implemented in scikit-learn (Pedregosa et al., 2011) as well as a multilayer perceptron (MLP) considering twenty tabular datasets, ten for regression and ten for classification.

The datasets are chosen based on popularity in the UCML repository (Kelly et al., 2019). We perform a two level cross-validation setup with 70% training, 15% validation, and 15% test splits. Hyperparameters were tuned on the validation set, and the models were retrained with the optimal parameters before final evaluation on the test set. Dataset details as well as the full data pipeline, pre-processing as well as hyperparameter selection and optimization details for all procedures are given in appendix A.6 together with code attached for reproducibility. In the hyperparameter search for CPD, we considered both the same ranks used for $(\text{MPO})^2$ and their squared values in order to obtain models with comparable parameter counts. During the hyperparameter search, we explored a broader range of CPD ranks, including both those used for $(\text{MPO})^2$ and their squared values to match parameter counts.

Due to the Hessian being unstable in the early phase of optimization, we applied Tikhonov regularization with an exponentially decaying schedule. To find the suitable regularization level we decay it and use early stopping to stop when validation loss does not decrease for 10 block/operator updates. In all tabular experiments we start with an initial value of $\lambda_{\text{start}} = 5$ and decay exponentially with $\gamma = 0.25$ as such: $\lambda_n = \lambda_{\text{start}} \cdot \gamma^n = 5.0 \cdot (0.25)^n$ where $n$ is the number of trained blocks and operators so far in the optimization order.

In Table 1 and Table 2 we report the average $R^2$ for regression and accuracy for classification, evaluated on the test set with the standard error of the mean over five random initializations of each model. In the upper part of each table we present the polynomial tensor network

based models, while the lower part contains generic function approximation models. The best overall model is highlighted in bold, and the best among the polynomial tensor network based methods is underlined. XGBoost was run in its default deterministic setting, and GP showed no variation in predictive performance across random seeds, so no error bars are reported for these two methods.

For both regression and classification, $(MPO)^2$ outperforms the other polynomial tensor network-based methods on most datasets, and when it is not the best, its performance remains close to the strongest alternative tensor based method. Overall, $(MPO)^2$ is on average the best performing tensor network based approach. Notably, feature ordering plays an important role when contrasting TNML-P and TNML-F with CPD and $(MPO)^2$. Since CPD and $(MPO)^2$ are invariant to feature ordering, they consistently outperform the MPS/TT structures that are feature order dependent.

Table 1: Models performance comparison for regression considering $R^2$ (higher is better).

|  | SP | AB | OB | BK | RE | EE | CO | AI | PO | SB | Avg |
|---|---|---|---|---|---|---|---|---|---|---|---|
| **$(MPO)^2$** | 21.69 ±0.12 | 58.97 ±0.35 | 73.53 ±0.55 | 74.23 ±0.13 | 80.82 ±0.71 | 99.78 ±0.01 | 85.40 ±0.44 | 42.45 ±0.37 | 1.96 ±0.24 | 60.84 ±2.48 | 59.97 |
| **CPD-A** | 20.93 ±0.28 | 59.82 ±0.07 | 71.18 ±2.04 | 73.65 ±0.21 | 81.26 ±0.84 | 99.69 ±0.01 | 80.89 ±0.79 | 40.44 ±0.56 | 1.71 ±0.03 | 68.72 ±2.92 | 59.83 |
| **CPD-S** | 19.60 ±2.37 | 56.19 ±0.75 | 51.86 ±0.49 | 41.45 ±0.12 | 10.25 ±5.19 | 92.65 ±0.12 | 49.36 ±5.81 | 31.06 ±1.68 | 1.13 ±0.17 | 44.26 ±5.38 | 39.78 |
| **TNML-P** | -700.45 ±190.59 | 55.69 ±0.68 | 48.80 ±2.51 | 61.61 ±0.26 | 78.10 ±0.22 | 99.64 ±0.02 | 73.27 ±1.69 | 27.91 ±0.03 | -9.06 ±0.07 | 20.31 ±4.49 | -24.42 |
| **TNML-F** | -1446.58 ±14.24 | -28.05 ±1.29 | -87.67 ±2.12 | -20.98 ±5.62 | -522.90 ±0.78 | -224.25 ±0.80 | -72.28 ±3.42 | 2.28 ±0.12 | -9.06 ±0.03 | -1676.04 ±2.84 | -408.55 |
| **GP** | **21.97** | **60.40** | **94.22** | – | 70.99 | 99.79 | 85.88 | – | – | – | **72.21** |
| **MLP** | 18.42 ±0.33 | 58.00 ±0.70 | 89.33 ±1.20 | 94.25 ±0.20 | 27.24 ±0.85 | 92.22 ±0.09 | 64.13 ±0.31 | **60.58** ±0.96 | **2.34** ±0.01 | 92.24 ±0.58 | 59.88 |
| **XGBoost** | 19.61 | 55.41 | 92.12 | **94.84** | **82.48** | **99.83** | **92.06** | 59.12 | 0.72 | **97.81** | 69.40 |

Table 2: Performance comparison of classification models (accuracy, higher is better).

|  | IR | HE | WQ | BR | AD | BA | WI | CE | SD | MU | Avg |
|---|---|---|---|---|---|---|---|---|---|---|---|
| **$(MPO)^2$** | **100.00** ±0.00 | **61.74** ±1.11 | 55.30 ±0.65 | **99.77** ±0.23 | 56.87 ±0.02 | 90.57 ±0.02 | **100.00** ±0.00 | **99.23** ±0.21 | 77.71 ±0.34 | 99.47 ±0.02 | 84.07 |
| **CPD-A** | **100.00** ±0.00 | 59.13 ±0.81 | 55.02 ±0.25 | 99.07 ±0.23 | 56.94 ±0.02 | 90.51 ±0.03 | 96.30 ±0.00 | 97.15 ±0.09 | 78.07 ±0.22 | 99.46 ±0.03 | 83.16 |
| **CPD-S** | **100.00** ±0.00 | 58.70 ±0.69 | 54.13 ±0.44 | 96.51 ±0.97 | 56.57 ±0.10 | 90.16 ±0.09 | **100.00** ±0.00 | 82.15 ±0.99 | 76.87 ±0.27 | 99.13 ±0.03 | 81.42 |
| **TNML-P** | **100.00** ±0.00 | 56.09 ±1.06 | 48.62 ±2.90 | 79.07 ±0.37 | 56.19 ±0.12 | 89.67 ±0.15 | 82.96 ±4.32 | 99.23 ±0.27 | 51.20 ±1.52 | 99.26 ±0.25 | 76.23 |
| **TNML-F** | 77.39 ±1.63 | 31.30 ±1.63 | 45.29 ±0.42 | 73.72 ±2.22 | 47.20 ±0.44 | 78.58 ±0.48 | 51.85 ±6.09 | 95.69 ±0.51 | 33.43 ±3.34 | 98.98 ±0.25 | 63.35 |
| **GP** | **100.00** – | 58.70 – | 61.03 – | 97.67 – | – | – | 96.30 – | 96.54 – | **78.31** – | – | 84.08 |
| **MLP** | 97.39 ±1.74 | 53.48 ±0.53 | 60.00 ±0.90 | 99.30 ±0.47 | 57.10 ±0.06 | **90.96** ±0.03 | 99.26 ±0.74 | 98.46 ±0.34 | 76.51 ±0.23 | 99.46 ±0.02 | 83.19 |
| **XGBoost** | **100.00** | 54.35 | **67.79** | 96.51 | **57.81** | 90.95 | **100.00** | 96.54 | 78.01 | **99.51** | **84.15** |

In Table 3 provided in supplementary section A.4, we systematically include an ablation study of the different modeling components of the $(MPO)^2$ procedure considering the Type I and Type II formulations (i.e., T1 and T2) as well as applications of the Masking (M) and Linear (L) MPOs. From the results we observe that all the specified $(MPO)^2$ variants produces best performance on at least one of the considered datasets. Consequently, the utility of the different $(MPO)^2$ variants are dataset dependent and the $(MPO)^2$ specification that is most suited for a given dataset needs to be accessed on the validation set. As such the versatile specification of the multivariate polynomial by the considered $(MPO)^2$ specifications

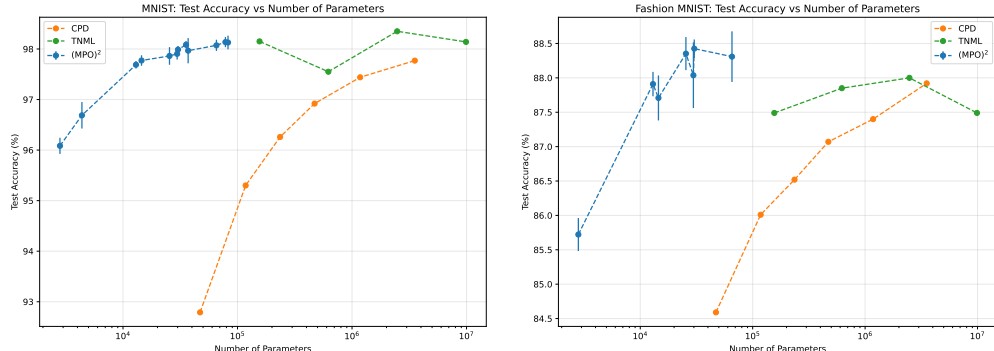

Figure 3: Accuracy on the test set for MNIST and Fashion MNIST classification tasks as function of parameters. CPD is optimized using the TeMPO procedure in Ayvaz & De Lathauwer (2022) as one-vs-all classifiers for each class. TNML results are reported from Efthymiou et al. (2019).

enable the systematic assessment of suitable tensor network specifications for multivariate polynomial regression with each dataset benefiting from different structures imposed.

We additionally report the average accuracy on the MNIST (Deng, 2012) and FashionM-NIST (Xiao et al., 2017) datasets in Figure 3, as a function of the number of parameters, comparing against the CPD Type I specification as this structure was imposed for these datasets in (Ayvaz & De Lathauwer, 2022) and TNML with the Fourier basis Efthymiou et al. (2019). Notably, for this image dataset we apply the Convolution MPO in our $(\text{MPO})^2$ procedure. Inspecting the Figure 3 we observe that the $(\text{MPO})^2$, can reach strong predictive performance using substantially fewer parameters, while reaching the same results as TNML, which due to the high number of blocks being feature dependent exhibit rapidly increasing parameters as function of ranks.

In the appendix, we furthermore explore exact polynomial inference, and devise an efficient inference procedure systematically growing the polynomial degree from lower degree learned $(\text{MPO})^2$ representations that naturally avoids overfitting when considering the modeling of noise-free polynomial functions in A.5.

## 5 CONCLUSIONS

We presented the $(\text{MPO})^2$ procedure for multivariate polynomial regression and demonstrated that this approach systematically outperformed conventional tensor network based polynomial regression modeling procedures based on existing CPD and MPS/TT based decompositions for polynomial regression. We attribute the enhanced performance to the $(\text{MPO})^2$ procedures higher modeling capacity when compared to CPD structures and feature order independence when comparing to existing MPS/TT based methodologies. Notably, we explored the versatility of the $(\text{MPO})^2$ framework leveraging Linear, Convolutional and Masking MPO to learn compressed feature representations and accounting for weight redundancies. We expect there are many further generalizations in which the MPO formalism can be used to accommodate other types of operations. As such, we also expect the $(\text{MPO})^2$ can be a useful tool when combined with deep learning modeling approaches akin to how the pi-sigma based CPD procedure has been imposed as nonlinear polynomial transformations of deep learning models Chrysos et al. (2022b); Panagakis et al. (2021); Chrysos et al. (2022a).

**Limitations** We presently only considered $(\text{MPO})^2$ modeling procedures in which the rank $R$ was specified to be identical across the MPO blocks. Future work should consider how individual ranks can be efficiently learned which would require an exponential evaluation of model specifications. It should also explore how Bayesian inference procedures can be used to quantify parameter uncertainty and automatically learn the relevance of different rank terms, see also Hinrich et al. (2020); Kilic & Batselier (2025).

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
