# OpenReview forum: "$(MPO)^2$: Multivariate Polynomial Optimization based on Matrix Product Operators"
_ICLR.cc/2026/Conference — ICLR 2026 Conference Withdrawn Submission_

### Official Review · Reviewer_Ur63 · 2025-10-22

**Soundness:** 1
**Presentation:** 1
**Contribution:** 1
**Rating:** 0
**Confidence:** 3

**Summary:**

In this paper, the authors propose $(\mathrm{MPO})^2$, a variant of the tensor networks. In this framework, the per-coordinate feature maps are replaced by matrix product operators (MPO). Compared with the usual  matrix product state (MPS) approach, this makes the model feature-order-independent and allows them to encode structured priors into the model more easily. The authors also conduct experiments on some dataset from the UCI-ML datasets.

**Strengths:**

It is nice for a model to be able to handle structured priors more easily and potentially beneficial to be feature-order-independent.

**Weaknesses:**

* The paper is poorly written. It is either rushed and/or machine translated, as there are numerous typos and confusing
  sentences in the paper, for example, the use of "apex" in Sec. 2.1 (maybe the authors mean "superscript") and line 184.
* I do not think the topic of this paper fits the scope of NeurIPS. None of the tensor network-based methods the authors
  compare in the experimental section are published in a machine learning conference/journal such as NeurIPS/ICML/JMLR.
  In addition, the method proposed by the authors cannot even beat more traditional learning methods such as Gaussian
  processes and XGBoost.
* Conceptually, I also do not see why we may want to use/study this method, as it neither gives strong empirical results
  nor is theoretically founded or amenable to theoretical analysis.

**Questions:**

See the Weakness section.

---

> ### Author Response · Authors · 2025-11-21
> **Answer part 1**
>
> We thank the reviewer for the constructive comments and careful assessment of our work. Below we address the concerns point by point.
>
> -----
>
> > The paper is poorly written. It is either rushed and/or machine translated, as there are numerous typos and confusing sentences in the paper, for example, the use of "apex" in Sec. 2.1 (maybe the authors mean "superscript") and line 184.
>
> We respectfully disagree with the assessment that the paper is poorly written and as stated by reviewer Q7Jc "This paper is quite clear and easy to understand."
>
> Regarding the use of "apex" this is clarified in line 108-110 of the manuscript where it is explained that "In this work, the position of the indices of a tensor, when referring to tensor networks structures, will be at the apex to indicate vertical modes in the diagrammatic representation and at the pedex to indicate horizontal modes." The apex use is thus to clarify the diagrammatical representations in Figure 2 and is related to the superscript notation as illustrated in Figure 1.
>
> If the reviewer finds other aspects of the paper and sentences confusing, please clarify what is found unclear, and we will be delighted to clarify any confusing aspects of the manuscript and improve the clarity of the paper as deemed necessary.
>
> > I do not think the topic of this paper fits the scope of NeurIPS. None of the tensor network-based methods the authors compare in the experimental section are published in a machine learning conference/journal such as NeurIPS/ICML/JMLR.
>
> The topic of the paper clearly falls within the scope of ICLR in particular covering the domains of supervised learning and representation learning which are central aspects of the ICLR conference calls for paper.
>
> We argue that tensor networks for polynomial regression is an underexplored topic within the community with clear merits that makes this an attractive area of further research, but we agree with the reviewer that this is a line of research presently underexplored in the NeurIPS/ICML/JMLR community.
>
> Recently, the community has adopted tensor network based methodologies for unsupervised learning, see also for instance:
>
> Glasser, Ivan, et al. "Expressive power of tensor-network factorizations for probabilistic modeling." Advances in neural information processing systems 32 (2019).
>
> Furthermore, tensor network structures have been used to compress deep learning represenations, see for instance:
>
> Kossaifi, Jean, et al. "Tensor regression networks." Journal of Machine Learning Research 21.123 (2020): 1-21.
>
> In addition, tensor network models have been used as building blocks within deep learning modeling frameworks, see for instance:
>
> Moulik Choraria, Leello Tadesse Dadi, Grigorios Chrysos, Julien Mairal, and Volkan Cevher. The spectral bias of polynomial neural networks. In International Conference on Learning Representations (ICLR), 2022.
>
> Wu, Yongtao, et al. Extrapolation and spectral bias of neural nets with hadamard product: a polynomial net study. NeurIPS 2022.
>
> and references therein.
>
> Finally, generic tools have been developed to accommodate the ML community to leverage tensor network based decompositions, see also for instance:
>
> Usvyatsov, Mikhail, Rafael Ballester-Ripoll, and Konrad Schindler. "tntorch: Tensor network learning with PyTorch." Journal of Machine Learning Research 23.208 (2022): 1-6.
>
> However, in the context of supervised learning we agree that modeling procedures purely relying on tensor network representations is a topic that has been given limited attention in the ML community. However, this methodology as we explain in the manuscript should have clear interest and benefits including:
> - The framework provides universal function approximation as an alternative to well established procedures including Gaussian Processes and Deep Learning.
> - The framework addresses the critical issue of curse-of-dimensionality by defining expressive models using compressed representations
> - Inference in these models can benefit from simple and easy to implement ALS-updates where sub-parts of the model are solved exactly conditioned on the rest of the state of the remaining system.
> - The framework produces easy interpretable symbolic equations associating the inputs to outputs producing a polynomial regression equation that facilitates explainable representations of model predictions.

---

> > ### Author Response · Authors · 2025-11-21
> > **Answer part 2**
> >
> > > In addition, the method proposed by the authors cannot even beat more traditional learning methods such as Gaussian processes and XGBoost.
> >
> > We would like to point out that we benchmark our approach specifically for the modeling of tabular datasets that contrary to large scale deep learning requires efficiently learning from limited number of observations. Furthermore, such tabular datasets while not part of large deep learning modeling challenges are still central to many machine learning tasks in which producing competitive predictions while preserving model interpretability is a central challenge. Notably, Gaussian Process and XGBoost as well as the MLP compared against we argue are the most prominent and strong learning frameworks for the considered problems while also enabling universal function approximations. Importantly, we here demonstrate that the simple (MPO)^2 framework in this context can produce competitive results while relying on simple polynomial regression equations for the predictions.
> >
> > > Conceptually, I also do not see why we may want to use/study this method, as it neither gives strong empirical results nor is theoretically founded or amenable to theoretical analysis.
> >
> > We respectfully also disagree with this statement. This line of research is important as it produces an alternative framework for universal function approximation (theoretically founded), admits simple ALS based inference (with convergence and properties well established and studied theoretically in the literature), and produces explainable representations in terms of symbolic regression providing simple mathematical expressions of the learned model predictions, also amenable to theoretical analysis. Importantly, we demonstrate that all these merits are possible while producing results that are comparable in performance to the state-of-the-art in the widely applied ML regime of making predictions from standard tabular datasets. Importantly, we also highlight the versatility of the framework by considering a series of relevant operators from linear projections and convolutions to masking. We apologize if we have failed to clearly communicate this in the manuscript and appreciate any feedback as to why the above does not make this a learning framework that is useful.

---

> > > ### Comment · Reviewer_Ur63 · 2025-11-23
> > >
> > > I thank the authors for the detailed response. I admit that I might be too harsh in my initial review and didn't understand parts of the manuscript as I'm not an expert in tensor networks. I will raise the score to 2, but, unfortunately, I couldn't further increase the score because I can't fully accept authors' justification on the relevance of the paper.
> > >
> > > I am in no position to judge if tensor networks are still relevant in 2025. However, I do not think the two papers the authors provided in the response  (Choraria et al, 2022; Wu et al, 2022) can justify the study of tensor networks, as both of them are still using some NTK-based justification in 2022, ignoring the long line of work since ~2020 that have proved NTK and/or other kernel-based methods cannot explain the success of deep learning. See, for example, [1-3].
> > >
> > > Second, having universal approximation property does not make a method theoretically founded. The main issue here is that, due to non-convexity of the training task, it is generally hard to find the solution that approximates the target. In fact, analyzing the non-convex training dynamics has been a central part of deep learning theory in the past 10 years.
> > >
> > > Third, I'm not sure if one can say being able to do symbolic regression implies interpretability. After all, understanding the behavior of multivariate polynomials is probably no easier than understanding a deep network.
> > >
> > > [1] Lenaic Chizat, Edouard Oyallon, Francis Bach. On Lazy Training in Differentiable Programming. 2018.
> > >
> > > [2] Behrooz Ghorbani, Song Mei, Theodor Misiakiewicz, Andrea Montanari. Linearized two-layers neural networks in high dimension. 2019.
> > >
> > > [3] Yuanzhi Li, Tengyu Ma, Hongyang R. Zhang. Learning Over-Parametrized Two-Layer Neural Networks beyond NTK. 2020.

---

### Official Review · Reviewer_Zbo6 · 2025-10-29

**Soundness:** 3
**Presentation:** 2
**Contribution:** 2
**Rating:** 4
**Confidence:** 2

**Summary:**

The paper introduces (MPO)²  for multivariate polynomial optimization based on Matrix Product Operator (MPO) structures. The approach integrates MPO-based polynomial weights with feature embeddings to pbtain a symmetry-preserving representation. Experimental results are reported on against existing tensor-decomposition-based polynomial regression methods on regression and classification tasks.

**Strengths:**

The paper introduces (MPO)², a new tensor-network-based architecture for modeling higher-order multivariate polynomials, extending conventional polynomial tensor networks.

By leveraging Matrix Product Operator (MPO), the model jointly learns feature and polynomial representations while maintaining feature-order invariance.

The method supports loss-agnostic optimization applicable to both regression and classification through second-order minimization.

**Weaknesses:**

The paper is not easy to follow, nor for readers to quickly get the main idea. I fel that the paper should carefully explain things woth more visually informative presentations to illustrate your idea.

The performance of the proposed model, when compared with existing baseliens, seems to possess limited advantages. For example, in regression tasks (Table~1), the proposed method is outperformed by XGBoost, GP. The average performance is very close to another tensor based method  - CPD-A. In classification tasks (table~2), the performance is again worse than XGboost and GP.

**Questions:**

(1) What is the index dd' in equation (5)?

(2) The number of coefficients  is claimed to be independent of the input features, and can you explicitly state what is the number of parameters?

(3) The authors should consider what is the key intended benefit of the proposed model if you are targeting on solving classification and regression tasks. for example, can you report the time consumption of tensor-based models or traditional ML algorithms to show if there is improvement in efficiency?

(4) If traditional baselines have higher accuracy in fitting the data, does it mean that their way of enforcing nonlinear correlation among input features are more advantageous?

---

> ### Author Response · Authors · 2025-11-24
> **Answer**
>
> We thank the reviewer for the constructive comments and careful assessment of our work. Below we address the concerns point by point.
>
> ---
>
> > What is the index dd' in equation (5)?
>
> It is representing a mode of the tensor, but the modes that in the graphical representation are vertical (acting in the distinct subspace of the inputs) are also denoted $d$ and the horizontal interactions over subspaces are denoted $r$.
>
> > The number of coefficients is claimed to be independent of the input features, and can you explicitly state what is the number of parameters?
>
> We would be grateful if the reviewer could point out where this is claimed, since this comes off as a mistake on our part. The number of parameters is dependent on the input features.
>
> The number of parameters of (MPO)^2 without any operator and with the same rank across blocks with rank $R$, input feature size $D$ and number of blocks is $(N-2) \cdot (R \cdot D \cdot R) + 2 \cdot R \cdot D$.
>
> Each block adds the product of its mode as number of parameter. Thus as an example, for a three legged block it would add rank squared by vertical leg dimension (associated usually with input features or with the compressed input features).
>
> The number of parameters is given by $dim(A)$, the product of the block's dimensions (i.e. a block with three legs, two ranks interaction and one input space interaction will have $dim(A)= R_A \times R_A \times D$).
>
> The linear operator has at worst blocks of 4 dimensions, (2 ranks, 2 vertical dimension) with the number of parameters $\cirq R^2 D D'$ times the number of blocks of the linear MPO.
>
> Convolutions have at worst a three dimensional block (2 ranks, 1 vertical dimension) with the number of parameters $\cirq R^2 D$, where ranks can differ between layers.
>
> The masking layer adds no additional parameters since the blocks are not learnable.
>
> > The authors should consider what is the key intended benefit of the proposed model if you are targeting on solving classification and regression tasks. for example, can you report the time consumption of tensor-based models or traditional ML algorithms to show if there is improvement in efficiency?
>
> The key benefit is being able to perform polynomial regression for high degree and leverage input transformations in a unified framework based on MPO structures, with the following merits:
>
> - The framework provides universal function approximation as alternative well established procedures including Gaussian Processes and Deep Learning.
>
> - The framework addresses the critical issue of curse-of-dimensionality by defining expressive models using compressed representations
>
> - Inference in these models can benefit from simple and easy to implement ALS-updates where sub-parts of the model are solved exactly conditioned on the rest of the state of the remaining system.
>
> - The framework produces easy interpretable symbolic equations associating the inputs to outputs producing a polynomial regression equation that facilitates explainable representations of model predi
>
> In [abalone_convergence_comparison_both.pdf] we provide a comparison between convergence times on the abalone dataset with (MPO)^2 with and without linear projection layer, and against CPD and MLP. Additionally, we provide convergence on MNIST in [train_test_comparison.png],
>
> > If traditional baselines have higher accuracy in fitting the data, does it mean that their way of enforcing nonlinear correlation among input features are more advantageous?
>
> The standard activation functions used in deep learning as non-linearities are basically infinite degree polynomials. For this reason, they can potentially represent the data better, at the cost of explainability. Polynomial models are used where explainability is needed to understand the interaction between the inputs.
>
> Contrasting other deep learning methods with polynomial methods, we would expect that deep learning is generally better in performance. The main advantage of polynomials is explainability and favoarable regularization by the compressed representations of the polynomial weights defined by the MPO^2 structure. Two things that are difficult to simultaneously achieve using general deep learning models. We focused on improving the polynomial regression formalism using tensor networks to avoid the curse of dimensionality that often critically hampers the use of polynomial regression.

---

> > ### Comment · Reviewer_Zbo6 · 2025-11-24
> > **Thanks for the reply**
> >
> > The number of parameters part - it was a typo, I was meant to ask what is the relation between input feature size and model size, and thanks for answering.
> >
> > Regarding the performance, I noted that my comment was not comparing your method with deep learning. In fact, the performance was not approaching those of XGBoost or  another tensor based method - CPD-A - in a number of cases.
> >
> > I am unable to open the attached PDF or PNG file, which I guess is not allowed.

---

> > > ### Author Response · Authors · 2025-11-25
> > > **Link for pdf**
> > >
> > > Yes, sorry about that I forgot to substitute with the link of the image
> > >
> > > [abalone_convergence_comparison_both](https://freeimage.host/i/abalone-convergence-comparison-both.f3xU4iG)
> > >
> > > [train_test_comparison](https://freeimage.host/i/train-test-comparison.f3x8fbn)

---

### Official Review · Reviewer_JCEu · 2025-10-31

**Soundness:** 2
**Presentation:** 2
**Contribution:** 2
**Rating:** 2
**Confidence:** 4

**Summary:**

This paper proposes (MPO)², a tensor network approach for multivariate polynomial optimization using Matrix Product Operators. The method addresses limitations of existing tensor decomposition approaches (CPD and TT) by providing feature-order independence and better handling of polynomial coefficient redundancies. The authors combine MPO polynomial weight tensors with MPO feature embeddings through three specialized operators: linear projection, convolution, and masking. Experiments on 20 small tabular datasets and MNIST/Fashion-MNIST show modest improvements over tensor-based baselines.

**Strengths:**

The mathematical development is rigorous and the MPO formulation elegantly addresses feature-order independence issues in tensor train methods. The specialized operators (linear, convolution, masking) represent thoughtful adaptations to different structural constraints.

The comparison includes relevant tensor network baselines and proper statistical evaluation with multiple random seeds across diverse small-scale problems.

The feature-order independence and explicit handling of coefficient symmetries address legitimate limitations in existing tensor decomposition approaches for polynomial regression.

The alternating natural gradient optimization and comprehensive ablation studies demonstrate careful experimental design within the chosen scope.

**Weaknesses:**

The paper provides no evidence that (MPO)² can scale to problems of practical modern relevance. Without parameter count comparisons to CNNs on MNIST/Fashion-MNIST, it's impossible to assess efficiency claims. The restriction to polynomial functions without nonlinear activations severely limits expressiveness compared to deep networks.

Evaluation is limited to toy datasets (mostly <10K samples, <60 features) that don't stress-test the method's claimed advantages. Modern deep learning routinely handles millions of parameters and features, while this work's largest dataset has 58 features. The absence of comparisons with modern neural architectures makes the practical relevance unclear.

No discussion of parameter scaling to larger problems, computational complexity comparisons with CNNs/MLPs, or analysis of why polynomial methods would be preferred over established deep learning approaches. The paper doesn't address whether (MPO)² could ever scale to billion-parameter regimes that are standard in current AI such as those LLMs (a 1B model is considered really small! which is already trained with several T tokens, definitely not do-able in (MPO)²).

The restriction to polynomial functions without nonlinearity severely constrains the method's applicability. Most modern ML problems require the expressive power of deep networks with nonlinear activations, which (MPO)² fundamentally cannot provide.

**Questions:**

1. Can you provide parameter count comparisons between (MPO)² and CNN/MLP baselines achieving similar accuracy on MNIST/Fashion-MNIST? How do parameter requirements scale with problem size?

2. What evidence suggests (MPO)² could scale to modern problem sizes (millions of features, billions of parameters)? What are the fundamental computational bottlenecks?

3. How does the restriction to polynomial functions limit practical applicability compared to neural networks with nonlinear activations? Can (MPO)² be extended with nonlinearities?

4. Given that billion-parameter models are now routine, what specific use cases justify choosing polynomial methods over established deep learning approaches?

5. How does the alternating natural gradient method's computational cost compare to standard gradient descent on equivalent-capacity neural networks?

6. Why not evaluate on high-dimensional problems where tensor methods should theoretically excel, rather than limiting to toy datasets where simple methods often suffice?

---

> ### Author Response · Authors · 2025-11-21
> **Answer to questions**
>
> We thank the reviewer for the constructive comments and careful assessment of our work. Below we address your concerns point by point.
>
> ---
>
> > Can you provide parameter count comparisons between (MPO)² and CNN/MLP baselines achieving similar accuracy on MNIST/Fashion-MNIST? How do parameter requirements scale with problem size?
>
> In the present manuscript, we contrast our approach to the existing polynomial regression based methodologies to highlight the parameter efficiency in comparison. However, we also agree that it is relevant to ground the results with respect to conventional deep learning based architectures and will in the revised manuscript and here provide the performance comparison with CNN and MLP as well.
>
> [fashion_mnist_accuracy_vs_params](https://freeimage.host/i/f3xeJlS)
>
> [mnist_accuracy_vs_params](https://freeimage.host/i/f3xkLjs)
>
> > What evidence suggests (MPO)² could scale to modern problem sizes (millions of features, billions of parameters)? What are the fundamental computational bottlenecks?
>
> Scaling, as in tackling the curse of dimensionality, is precisely the idea behind using tensor networks decompositions, where ideally we would want to represent those same billions of parameters and features with the simplest possible tensor network structure. However, when dealing with structures with constrained subspaces (the blocks), scaling might cause performance loss due to degeneracy of the parameters.
>
> In order to accommodate a higher number of features, it is possible to split the features in subspaces and elongate the model by assigning subsequent smaller blocks to each subspace, or by applying a convolution-like operator over the new subspaces.
>
> The high dimensionality solved by this class of tensor networks is the one that naturally appears when dealing with high degree and high dimension polynomials. Consequently, the dimensionality that these models are compressing is on the space created by the tensor product of the inputs, not the high dimensionality of the inputs itself.
>
> In our case, the main bottleneck for scaling to higher orders is definitely in solving the linear system to find the block update.
>
> Several approaches are more scalable here than the presently used natural gradient update. However, we find that highly scalable gradient based methodologies suffer from poor convergence whereas intermediate scalable approaches such as conjugate gradient descent are also hampered in their convergence and less stable.
>
> See [train_test_comparison](https://freeimage.host/i/f3x8fbn) where we compare standard CNN with AdamW gradient descent against the (MPO)^2 model with full parameter gradient descent, block-wise gradient descent (five epochs per block in alternating fashion), and the standard natural gradient update. We see that the natural gradient method obtains a faster convergence and higher test accuracy compared with the slower convergence and lower performance of the standard gradient methods.
>
> In the revised manuscript we have added a discussion on scaling in the discussion section.

---

> > ### Author Response · Authors · 2025-11-21
> > **Answer part 2**
> >
> > > How does the restriction to polynomial functions limit practical applicability compared to neural networks with nonlinear activations? Can (MPO)² be extended with nonlinearities?
> >
> > As for the practical applicability, this is highly dependent on the context. In general cases non-linearities in deep learning based methods are de facto representing infinite degree polynomials, as any non-linearity can be seen as a Taylor series. Adding generic non-linearities destroys the polynomial nature of the model in the sense that it is not defined by the polynomial degree anymore and by how the inputs interact with each other. And as such, causing the model to be a black box, where in contrast (MPO)² is explainable due to its polynomial nature. However, (MPO)² can indeed be extended by simply applying a non-linear transformation over the space of ranks and then contracting. In this way, the model would look very similar to an MLP.
> >
> > We would also assume that high-dimensional machine learning problems such as in LLMs or computer vision models would not benefit by being represented end-to-end by a polynomial, which could not replace the complexity of fully stacked deep learning models. However, a subpart of the model could be useful to be represented as a polynomial, like a polynomial attention layer where it becomes interesting to evaluate the strength of interactions, or using multiple stacks of low degree polynomials represented by MPOs connected or combined with activations function. Replacing the attention mechanism with such a polynomial tensor network is a promising direction for future work.
> >
> > Importantly, we believe that we found a parameter efficient and high-fidelity way of representing polynomials with tensor networks overcoming the curse of dimensionality. If used in an end-to-end model they have the same capacity of representation as polynomials and the same benefits, so any problem where polynomial regression is relevant is also immediately relevant for our proposed model.
> >
> > We believe that general deep learning structures could benefit from introducing polynomial interactions, and it would be interesting to investigate such models equipped with a (MPO)² layer capturing higher-order interactions.
> >
> > > Given that billion-parameter models are now routine, what specific use cases justify choosing polynomial methods over established deep learning approaches?
> >
> > Polynomial methods are often used when not only the prediction is relevant, but also the understanding of the underlying interaction. We would say that whenever your problem requires an understanding of the strength of interactions between variables. Problems that span from economy, physics and chemistry use polynomial regression.  Importantly, polynomial regression provides explicit easy interpretable symbolic equations whereas deep learning approaches lack explainability.
> >
> > Additionally, billion-parameter models are only routine in computer vision and natural language processing. Other domains still routinely use much smaller models, with parameter counts in the millions not in the billions, that are finetuned to a particular domain and use domain-specific inductive biases and which outperform foundation models in their specific domain. Our work introduces a new class of models to the general deep learning literature and not specifically targeting or competing against foundation models.
> >
> > > How does the alternating natural gradient method's computational cost compare to standard gradient descent on equivalent-capacity neural networks?
> >
> > We will provide a comparison on MNIST and on the abalone dataset.
> >
> > In [train_test_comparison](https://freeimage.host/i/f3x8fbn) we compare the convergence time between models of the same size. Here you can see how the natural gradient update is both faster to converge and converges at a higher value.
> >
> > Additionally, we provide convergences across model sizes that were used on the tabular data. In [abalone_convergence_comparison_both](https://freeimage.host/i/f3xU4iG) we provide the convergence times for similar sized models on the abalone dataset. Note that the natural gradient update with (MPO)² obtains the lowest training loss.

---

> > > ### Author Response · Authors · 2025-11-21
> > > **Answer part 3**
> > >
> > > > Why not evaluate on high-dimensional problems where tensor methods should theoretically excel, rather than limiting to toy datasets where simple methods often suffice?
> > >
> > > In our setting, the tensor network factorization operates over polynomial degrees rather than across feature modes. A single block represents all first-order terms, a second block represents all second-order interactions across all features, and so on. The compression benefits arise primarily from the structure of interactions rather than from the feature dimension itself. For this reason, although the method can be extended to high-dimensional inputs, this is not where the current formulation offers its strongest advantage.
> > >
> > > Scaling to very high-dimensional data was infeasible due to the memory cost of natural gradient updates in our current implementation. Reducing this burden will require replacing or approximating the natural gradient with lighter first order optimization methods. Developing such training strategies is an important direction for future work, since it would allow the model to operate in higher dimensional settings without prohibitive memory usage.
> > >
> > > If such scalability is achieved, then the model can benefit from more structured input representations. Assigning an MPO to each mode of a high-dimensional tensor such as height, width, and channels, rather than patchifying as done currently, would exploit the underlying structure of the data more effectively. This provides a clear path for extending the method to genuinely high-dimensional problems.
> > >
> > > It would also be valuable to investigate integration of our model as a modular component within neural architectures. For instance, a version of our (MPO)² model with three blocks and the linear projection operator aligns closely with the structure of self-attention, where the three blocks correspond to Q, K, and V and compute higher order interactions. This makes it a plausible drop-in alternative to attention mechanisms.
> > >
> > > Our primary goal in this work was to demonstrate competitive performance within polynomial tensor decomposition methods and to close the gap to existing universal function approximation approaches for tabular data. The advantages of explicit polynomial structure remain especially relevant in settings where interpretability is important.

---

> > > > ### Comment · Reviewer_JCEu · 2025-11-24
> > > > **Thanks for your reply**
> > > >
> > > > Thanks for your additional experiments and feedback. It is seen that CNN, at the same parameter count, still performs better than MPO2 even for relatively simple CV tasks.
> > > >
> > > > "so any problem where polynomial regression is relevant is also immediately relevant for our proposed model."<-- this I agree, but then it's like proposing MPO2 and then trying in retrospect to find problems where MPO2 excels.
> > > >
> > > > "a subpart of the model could be useful to be represented as a polynomial, like a polynomial attention layer where it becomes interesting to evaluate the strength of interactions, or using multiple stacks of low degree polynomials represented by MPOs connected or combined with activations function." <-- Also, scaling up MPO2 isn't practical, but then infusing MPO2 into layers of a DNN would destroy the original explainability intent of MPO2.
> > > >
> > > > Given these, I'm afraid I'll maintain my original scores.

---

### Official Review · Reviewer_Q7Jc · 2025-11-05

**Soundness:** 2
**Presentation:** 2
**Contribution:** 2
**Rating:** 2
**Confidence:** 4

**Summary:**

This paper proposes (MPO)^2, a new method for multivariate polynomial optimization that solves the "curse of dimensionality", i.e., the exponential growth of polynomial coefficients. Unlike prior tensor-based methods like CPD (which is less expressive) or TT/MPS (which are dependent on the order of input features), (MPO)^2 uses Matrix Product Operators to create a compact, highly expressive, and feature-order independent model. On various regression and classification tasks, (MPO)^2 demonstrated superior performance compared to these existing approaches.

**Strengths:**

1. This paper extends the CP/TT-structure multivariate polynomial optimization framework to a two-layer MPO structure. Specifically, linear projection, convolution, and masking MPO structures are discussed in the paper. The experimental results also demonstrate its superiority.

2. This paper is quite clear and easy to understand.

**Weaknesses:**

1. The novelty of this paper is limited as it extends the CP-structured multivariate polynomial optimization to the MPO structure with a general linear mapping. Thus, the main advantages and novelty seem to be limited.

2. The main advantages of TN-based multivariate polynomials are also not clear. In the experimental section, only limited baselines like MLP, GP, etc are compared in two experiments. In addition, what kind of mapping did the author use for the two experiments? Linear projection? Convolution MPO? or Masking MPO?

3. The motivation of this paper is also unclear.  In Section 2.2.1, the authors claimed that "a linear transformation is ... suitable latent feature ..." is not a very strong motivation for this method. If so, why not stack the MPO structure into a three-layer or even N-layer? Compared with the one-layer CP/TT structure, the main motivation for proposing this (MPO)^2 has not been clearly illustrated.

4. In the optimization algorithm, the authors adopted the natural gradient descent for optimizing the core tensors of MPO. Why not directly adopt the SGD or Adam-like first-order optimization algorithm? Natural gradient descent requires computing the Fisher information matrix,  which will cause high computation cost when the core tensor A is relatively large.

5. The computational time should be discussed or presented, as the two-layer structure may dramatically increase the computational cost.

**Questions:**

See the above weakness section.

---

> ### Author Response · Authors · 2025-11-21
> **Answer part 1**
>
> We thank the reviewer for the constructive comments and careful assessment of our work. Below we address the concerns point by point.
>
> -----
>
> > The novelty of this paper is limited as it extends the CP-structured multivariate polynomial optimization to the MPO structure with a general linear mapping. Thus, the main advantages and novelty seem to be limited.
>
> We believe that the novelty of this paper with respect to the existing CP-structured multivariate polynomial regression, is the introduction of the MPOs, substituting the role of the CP structure, not mapping from it. MPOs and the CP structure are two radically different decompositions with different properties and span of applications.
>
> Feature based tensor train/MPS and MPO methods for machine learning, as far as we know have always been structured around the feature dimension (A block / subspace for each feature) and they did not explore the feature invariant formalism that we propose.
>
> Additional novelty is that we can model linear operation as a whole using MPOs.
>
> > The main advantages of TN-based multivariate polynomials are also not clear. In the experimental section, only limited baselines like MLP, GP, etc are compared in two experiments. In addition, what kind of mapping did the author use for the two experiments? Linear projection? Convolution MPO? or Masking MPO?
>
> What we show in the table was the best performing model through different MPOs, as it is highlighted in the ablation studies in the supplementary material, where we present all results of all type of structures with their associate performances, see supplementary section A.4.
>
> The results we found show that our model works best in the field of polynomial models. And the study in the appendix further demonstrates how we get a good approximation of the exact polynomial coefficients.
>
> Contrasting other deep learning methods with polynomial methods, we would expect that deep learning is generally better in performance. The main advantage of polynomials is explainability and regularization by the induced structure imposed on the polynomial coefficients by the (MPO)^2. Two aspects that are generally very difficult to simultaneously achieve using deep learning modelling approaches.
>
> We believe that the advantage of using the MPO method for representing polynomials is shown by comparing the performance and number of parameters against other polynomial methods.
>
> The performance difference between polynomial regression and other machine learning / deep learning methods is very problem dependent. However, polynomial regression produces symbolic and interpretable mathematical equations associating inputs with outputs particularly useful when explainability and the understanding of feature interactions are important. Furthermore, polynomial regression is still used in various fields including economy and physics.

---

> > ### Author Response · Authors · 2025-11-21
> > **Answer part 2**
> >
> > > The motivation of this paper is also unclear. In Section 2.2.1, the authors claimed that "a linear transformation is ... suitable latent feature ..." is not a very strong motivation for this method. If so, why not stack the MPO structure into a three-layer or even N-layer? Compared with the one-layer CP/TT structure, the main motivation for proposing this (MPO)^2 has not been clearly illustrated.
> >
> > We wish to stress that the method's novelty is not confined to adding a linear transformation, but constructing the full problem of polynomial representation and learning in an MPO space. Importantly, our framework provides a feature invariant model, introducing in this framework the freedom of modeling linear transformation over the fully expanded polynomial space, which is often impractical due to the size of the space and naturally embedding the linear transformations in an MPO structure.
> > Standard tensor train (TT) methods often decompose the space, assigning a block to each of the features, while our formulation consider a block for each degree of interaction, providing better results to the standard feature based TT approach. Also existing works often rely on ALS or DMRG that are learning method specific for squared losses, while we extend our algorithm to incorporate any differentiable loss. Additionally, we introduce stacking that can solve problems including degeneracy of coefficients and feature compression.
> >
> > It is definitely possible to stack MPOs, and it would bring the same benefits as stacking linear transformation to the inputs. Often, and for the space transformation we wished to apply, one layer was enough. The degeneracy-removing mask operator would not provide anything if stacked. It is however possible that complex operations over the input might need to be modeled by multiple layers of MPOs. Three subsequent compressions of the subspace, modeled by a product of three matrices is equivalent to one compression of the space. That is why vertical stacking of pure compressing operators can harm the performance. Notably, in the mask case the operator does not compress but applies an operation on the input that is achieved with only one layer.
> >
> > We point out that in this model, differently from standard procedures, a vertical stacking does not inrease the capability and complexity of the model, but a horizontal expansion does; increasing the effective degree of the polynomial and consequently the capacity of the model.
> >
> > > In the optimization algorithm, the authors adopted the natural gradient descent for optimizing the core tensors of MPO. Why not directly adopt the SGD or Adam-like first-order optimization algorithm? Natural gradient descent requires computing the Fisher information matrix, which will cause high computation cost when the core tensor A is relatively large.
> >
> > Yes, the problem suffers from computational constraint due to the Hessian size. This is a problem that surfaces whenever adopting tensor networks structure in any type of second order algorithm (DMRG and ALS).
> >
> > To see the difference in performance and computational costs we provide MNIST classification accuracy as a function of training time with natural gradient optimization, standard AdamW with full model parameter gradient descent, block-wise AdamW gradient descent (five epochs per block), and finally comparing with the convergence of a standard CNN trained with AdamW in gradient descent. See [train_test_comparison](https://freeimage.host/i/f3x8fbn) for these results. The natural gradient method achieves a higher performance and a faster convergence, but interestingly the full parameter AdamW model also obtained comparable performance with the CNN model. This is in contrast to traditional tensor network optimization literature and is a promising future direction for scaling models to higher-dimensional problems.

---

> > > ### Author Response · Authors · 2025-11-21
> > > **Answer part 3**
> > >
> > > > The computational time should be discussed or presented, as the two-layer structure may dramatically increase the computational cost.
> > >
> > > The dominant part of the computational complexity is solving the Hessian linear system for the natural gradient update. Solving a linear system for a $n\times n$ matrix costs at most $n^3$. The Hessian of a block has size $dim(A)\times dim(A)$, where $dim(A)$ is the product of the block's dimensions (i.e. a block with three legs, two ranks interaction and one input space interaction will have $dim(A)= R_A \times R_A \times D$) and the total cost will be $dim(A)^3$. So adding a new layer depends entirely on the blocks that form it, and whether the blocks are learnable.
> > >
> > > The linear operator has at worst blocks of 4 dimensions, (2 ranks, 2 vertical dimension) adding a computational cost of order $R^6 D^3 D'^3$. Note that often the ranks in the blocks of this operator are $1$ or very small.
> > >
> > > Convolutions have at worst a three dimensional block (2 ranks, 1 vertical dimension), so the additional complexity is of the order of $R^6 D^3$ where ranks can differ between layers.
> > >
> > > The masking layer adds negligible computation since the blocks are not learnable.
> > >
> > > In [abalone_convergence_comparison_both](https://freeimage.host/i/f3xU4iG), we provide convergence times for models of similarly varying sizes, including between (MPO)^2 without any extra layer and (MPO)^2 with the linear projection layer.
> > >
> > > In summary, adding blocks increases the computational time linearly, the bottleneck of computation depends on the block sizes. In the revised manuscript we will carefully discuss these computational aspects.

---

> ### Comment · Reviewer_Q7Jc · 2025-11-25
>
> I appreciate the authors' detailed rebuttal to my concerns. Part of my concerns, including the computational concerns, have been solved. However, the novelty and the motivation for this approach have not reached the borderline of ICLR, therefore, I would maintain my original score.

---

### Note · Authors · 2025-12-03

**Comment:**

We sincerely thank the reviewers for the detailed and constructive feedback on our submission. While we are excited about the direction of this work, we acknowledge the limitations also pointed out during the review.
We have decided to withdraw the paper to further implement and compare scalable variants.
We are grateful for the evaluations and will focus on incorporating the feedback to improve our work.

**Withdrawal Confirmation:**

I have read and agree with the venue's withdrawal policy on behalf of myself and my co-authors.